# Altered subgenomic RNA abundance provides unique insight into SARS-CoV-2 B.1.1.7/Alpha variant infections

Matthew D. Parker [1,2], Hazel Stewart [3], Ola M. Shehata[4], Benjamin B. Lindsey [5,6], Dhruv R. Shah [6], Sharon Hsu[2,6], Alexander J. Keeley [5,6], David G. Partridge [5], Shay Leary[7], Alison Cope[5], Amy State[5], Katie Johnson[5], Nasar Ali[5], Rasha Raghei[5], Joe Heffer [8], Nikki Smith[6], Peijun Zhang[6], Marta Gallis[6], Stavroula F. Louka [6], Hailey R. Hornsby [6], Hatoon Alamri[4], Max Whiteley[6], Benjamin H. Foulkes[6], Stella Christou [6], Paige Wolverson [6], Manoj Pohare [6], Samantha E. Hansford [6], Luke R. Green [6], Cariad Evans[5], Mohammad Raza[5], Dennis Wang [1,2,9], Andrew E. Firth[3], James R. Edgar [3], Silvana Gaudieri[9,10,11], Simon Mallal[10,11], The COVID-19 Genomics UK (COG-UK) consortium*, Mark O. Collins [4], Andrew A. Peden [4] & Thushan I. de Silva [5,6✉]

B.1.1.7 lineage SARS-CoV-2 is more transmissible, leads to greater clinical severity, and results in modest reductions in antibody neutralization. Subgenomic RNA (sgRNA) is produced by discontinuous transcription of the SARS-CoV-2 genome. Applying our tool (periscope) to ARTIC Network Oxford Nanopore Technologies genomic sequencing data from 4400 SARS-CoV-2 positive clinical samples, we show that normalised sgRNA is significantly increased in B.1.1.7 (alpha) infections ($n = 879$). This increase is seen over the previous dominant lineage in the UK, B.1.177 ($n = 943$), which is independent of genomic reads, E cycle threshold and days since symptom onset at sampling. A noncanonical sgRNA which could represent ORF9b is found in 98.4% of B.1.1.7 SARS-CoV-2 infections compared with only 13.8% of other lineages, with a 16-fold increase in median sgRNA abundance. We demonstrate that ORF9b protein levels are increased 6-fold in B.1.1.7 compared to a B lineage virus in vitro. We hypothesise that increased ORF9b in B.1.1.7 is a direct consequence of a triple nucleotide mutation in nucleocapsid (28280:GAT > CAT, D3L) creating a transcription regulatory-like sequence complementary to a region 3' of the genomic leader. These findings provide a unique insight into the biology of B.1.1.7 and support monitoring of sgRNA profiles to evaluate emerging potential variants of concern.

[1] Sheffield Biomedical Research Centre, The University of Sheffield, Sheffield, UK. [2] Sheffield Bioinformatics Core, The University of Sheffield, Sheffield, UK. [3] Department of Pathology, University of Cambridge, Cambridge, UK. [4] Department of Biomedical Science, The University of Sheffield, Western Bank, Sheffield, UK. [5] Sheffield Teaching Hospitals NHS Foundation Trust, Sheffield, UK. [6] The Florey Institute for Host-Pathogen Interactions & Department of Infection, Immunity and Cardiovascular Disease, Medical School, University of Sheffield, Sheffield, UK. [7] Institute for Immunology and Infectious Diseases, Murdoch University, Murdoch, WA, Australia. [8] IT Services, The University of Sheffield, Sheffield, UK. [9] Department of Computer Science, The University of Sheffield, Sheffield, UK. [10] Division of Infectious Diseases, Department of Medicine, Vanderbilt University Medical Center, Nashville, TN, USA. [11] School of Human Sciences, University of Western Australia, Crawley, WA, Australia. *A list of authors and their affiliations appears at the end of the paper.
✉email: t.desilva@sheffield.ac.uk

The SARS-CoV-2 lineage B.1.1.7 (Alpha, 20I/501Y.V1)[1] has been classified as a variant of concern by public health agencies. An increasing body of evidence suggests B.1.1.7 is more transmissible[2,3] and rapidly became the dominant circulating virus in the United Kingdom (UK) during October 2020 to February 2021 (Fig. 1a). To date, B.1.1.7 has been reported in 150 countries (https://cov-lineages.org/, 4 August 2021), with increasing prevalence. Preliminary data[4,5] shows that B.1.1.7 positive diagnostic respiratory samples may have lower cycle

threshold (Ct) values, therefore higher viral loads, compared to other lineages. These findings suggest a potential reason for enhanced transmissibility, though they did not account for potential confounders such as days since symptom onset at sampling. Many of these studies also use S gene target failure (SGTF) as a surrogate for the presence of B.1.1.7[4], which might misclassify samples, depending on the prevalence of B.1.1.7[6]. Several analyses from community-tested cases also suggest increased mortality associated with B.1.1.7[6]. Reasons for the

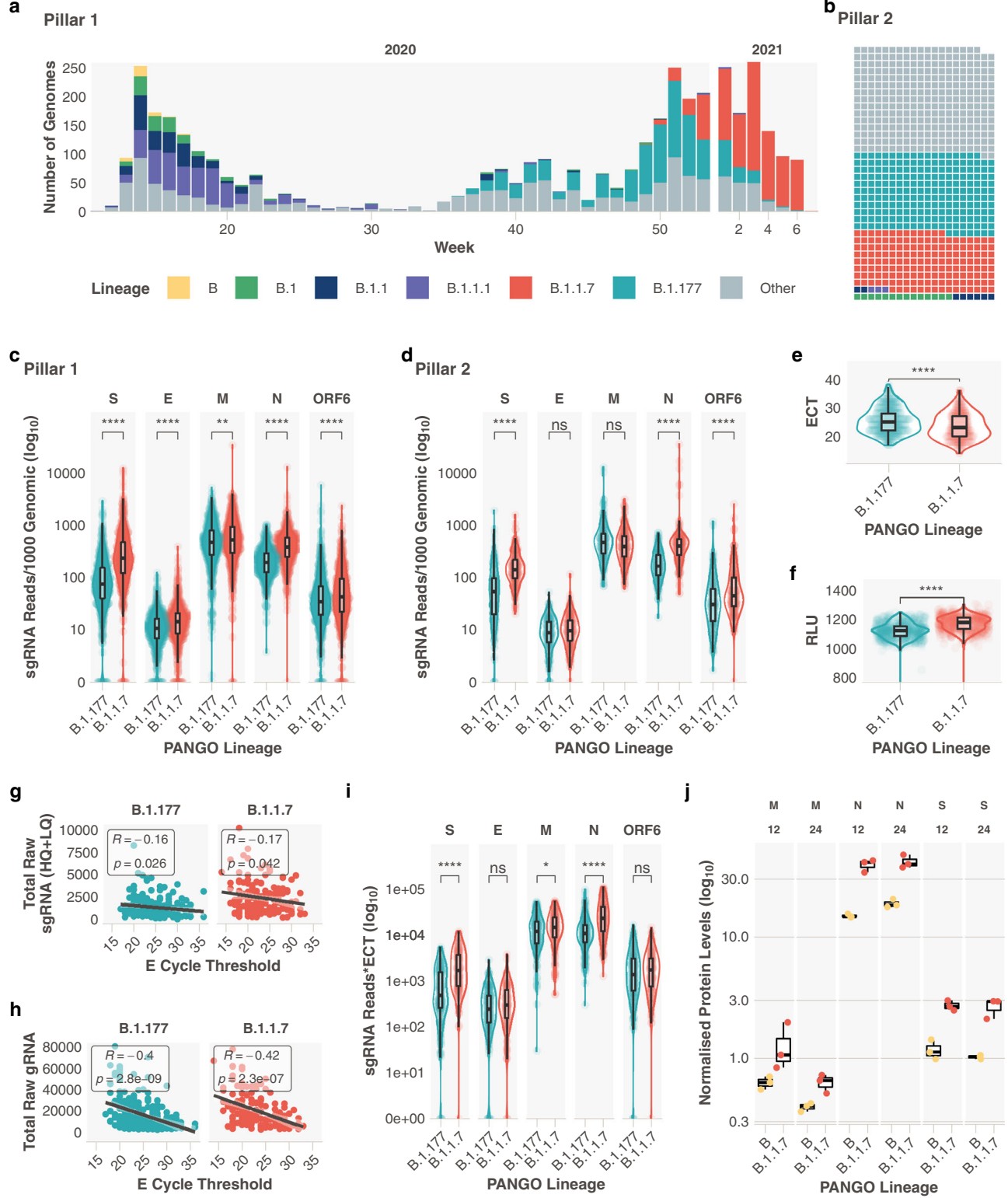

**Fig. 1 Subgenomic RNA abundance is increased in B.1.1.7 infections. a** Pillar 1 SARS-CoV-2 lineages over time (B.1.1.7$_n$ = 729, B.1.177$_n$ = 764). The number of genomes is representative of the number of positive cases in pillar 1 at that time, which reflects the epidemic curves of the SARS-CoV-2 pandemic in the UK. **b** Representation of lineage composition of pillar 2 data (B.1.1.7$_n$ = 150, B.1.177$_n$ = 179). **c** sgRNA abundance in samples of lineages B.1.177 and B.1.1.7 in pillar 1 samples from the most highly expressed ORFs. **d** sgRNA abundance in samples of lineages B.1.177 and B.1.1.7 in pillar 2 samples from the most highly expressed ORFs. **e** E gene cycle threshold (ECT) for B.1.177 (n = 257) and B.1.1.7 (n = 185) lineages. **f** Relative light units (RLU) for B.1.177 (n = 626) and B.1.1.7 (n = 626) lineages. **g** E gene cycle threshold compared to total raw sgRNA counts (high and low quality) for B.1.177 and B.1.1.7. Correlation coefficient and p value using Pearson. **h** E gene cycle threshold compared to total raw genomic RNA counts for B.1.177 and B.1.1.7. Correlation coefficient and p value using Pearson. **i** Raw sgRNA counts normalised to E gene cycle threshold (sgRNA*ECT). **j** Protein abundance for S M and N proteins measured by LC-MS/MS and normalised to ORF1a in B and B.1.1.7 infected TMPRSS2 & ACE2 expressing A549 cells at 12 and 24 h post infection (n = 3 independent experiments). All p values (except **g**, **h**, **j**) calculated using an unpaired Wilcoxon signed-rank test and adjusted for multiple testing with the Holm method (**** <0.0001, *** <0.001, ** <0.01, * <0.05). All boxplots depict the 25th, 50th (median), and 75th percentiles, and whiskers represent the most extreme datapoint which is no more than 1.5x the interquartile range. B.1.1.7 is represented in red, B.1.177 in teal, B in yellow, B.1 in green, B.1.1 in navy blue, B.1.1.1 in purple and other lineages in grey.

potential viral load increase and enhanced mortality are currently unclear[7].

Genomic surveillance has been critical in rapidly identifying these variants and Nanopore sequencing of ARTIC Network[8] prepared SARS-CoV-2 amplicons is used by many laboratories to generate this data. We have reported an approach to quantify subgenomic RNA (sgRNA) abundance in genomic sequence data, which is produced as a result of a critical step in the SARS-CoV-2 replication cycle[9]. sgRNA is produced from the genomically encoded SARS-CoV-2 RNA-dependent RNA polymerase (RdRp) using discontinuous transcription of the positive, single-stranded SARS-CoV-2 genome from the 3′ end. Negatively stranded RNAs are produced, which are shorter than the genome, owing to a template switch from the ORF to the leader sequence at the 5′ end of the genome when RdRp encounters a transcription regulatory sequence in the genome body (TRS-B) to a complementary TRS 3′ of the leader sequence (TRS-L). All sgRNAs, therefore, contain a leader sequence at their 3′ end which can be used computationally for their identification. There are thought to be nine such canonical sgRNAs; Spike:S, E: Envelope, M: Membrane, N: Nucleocapsid, ORF3a, ORF6, ORF7a, ORF8 and ORF10, although multiple studies have found negligible ORF10 expression[10,11].

As part of COVID-19 Genomics Consortium UK (COG-UK)[12] we have sequenced SARS-CoV-2 positive combined nose and throat swabs from healthcare workers and patients at Sheffield Teaching Hospitals NHS Foundation Trust, Sheffield, UK ('Pillar 1' testing, Supplementary Data 1). Additionally, to relieve pressure on centralised sequencing services, we have also sequenced a selection from 'Pillar 2' testing, which represents SARS-CoV-2 positive samples from the community, tested at the UK's lighthouse laboratories (Supplementary Data 2).

We hypothesised that we would see differences in sgRNA abundance in distinct lineages of SARS-CoV-2, in particular, increased sgRNA in B.1.1.7 that may relate to its altered clinical phenotype. We find that normalised sgRNA is significantly increased in B.1.1.7 infections when compared to infections caused by B.1.177, the previously dominant circulating lineage in the UK. We also demonstrate that a noncanonical sgRNA which likely represents ORF9b is found in most B.1.1.7 infections and that ORF9b protein levels are increased significantly in B.1.1.7 compared to B lineage viruses during in vitro culture.

## Results and discussion

**Subgenomic RNA abundance in B.1.1.7 compared to other lineages**. We stratified sgRNA abundance by lineage in 4400 SARS-CoV-2 sequences that reached our previously defined quality control thresholds (>90% genome coverage, >50 K mapped reads, Supplementary Table 2 and Supplementary Data Files 3, 4), normalised for the genomic coverage from the corresponding amplicon for the five most abundantly expressed open reading frames (S, E, M, N and ORF6). We primarily compared B.1.1.7 with B.1.177 (Supplementary Table 1), the previously dominant lineage in the UK. These two lineages are also the most represented sequences from both sampling pillars (Supplementary Fig. 1 and Supplementary Table 2) in our dataset (Fig. 1a, b, B.1.1.7; Pillar 1: 729, Pillar 2: 150, B.1.177; Pillar 1: 764 Pillar 2: 179). A statistically significant increase in normalised abundance of sgRNA for S, N, and to a lesser extent E, ORF6 and M, was apparent in B.1.1.7 SARS-CoV-2 infections (Fig. 1c, d, Wilcoxon effect sizes; S: 0.475, E: 0.191, M 0.0700, N: 0.469, ORF6: 0.105). Negligible differences were seen in other ORF sgRNA (Supplementary Fig. 2). Consistent with previous findings[4,13], we also found significantly decreased E gene Ct (ECT; in house diagnostic assay;[14] B.1.177 median = 25, B.1.1.7 median = 23) and significantly greater Relative Light Units (RLU; Hologic Panther platform; B.1.177 median = 1121, B.1.1.7 median = 1177) in B.1.1.7 infections compared to B.1.177 infections (Fig. 1e, f). Accordingly, total mapped reads and total genomic RNA reads (i.e. non-sgRNA) were also significantly higher for B.1.1.7 sequences compared to B.1.177 (Supplementary Fig. 3). To rule out any confounding effects of our normalisation method, we also normalised subgenomic RNA reads to reads from ORF1a. Since ORF1a RNA is produced entirely from replication there would be no contribution of nested subgenomic RNA to the read count. Normalising to ORF1a reveals the same pattern of significant overrepresentation of S, N, E and M sgRNA in B.1.1.7 (Supplementary Fig. 4).

We observed a weak but statistically significant negative correlation between raw sgRNA reads and ECT ($R = -0.16$ and $p = 0.026$ for B.1.177, $R = -0.17$ and $p = 0.042$ for B.1.1.7), with a greater negative correlation between genomic reads and ECT ($R = -0.4$ and $p < 0.0001$ for B.1.177, $R = -0.42$ and $p < 0.0001$ for B.1.1.7; Fig. 1g, h). To ensure the observed increase in sgRNA abundance in B.1.1.7 was not due to the effect of ECT alone (as a surrogate of viral load), we normalised sgRNA to ECT and repeated our comparisons. The differences between B.1.1.7 and B.1.177 were still apparent for S, N and M sgRNA (Fig. 1i), suggesting that the increase in B.1.1.7 sgRNA abundance is seen independently of any difference in ECT between lineages.

To confirm subgenomic RNA abundance increases in B.1.1.7 were translated to an increase in protein levels we performed quantitative mass spectrometry on *ACE2, TMPRSS2* expressing A549 cells infected (MOI = 1) with either B or B.1.1.7 SARS-CoV-2 isolates. At 12 and 24 h post infection, cell monolayers were lysed and protein lysates were subsequently analysed by LC-MS/MS analysis. In order to quantify differences in the abundance of SARS-CoV-2 proteins between strains and at different time points, label-free quantification (LFQ) data were normalised to the abundance of proteins encoded by the ORF1ab gene (Nsp1, Nsp2 and Nsp3).

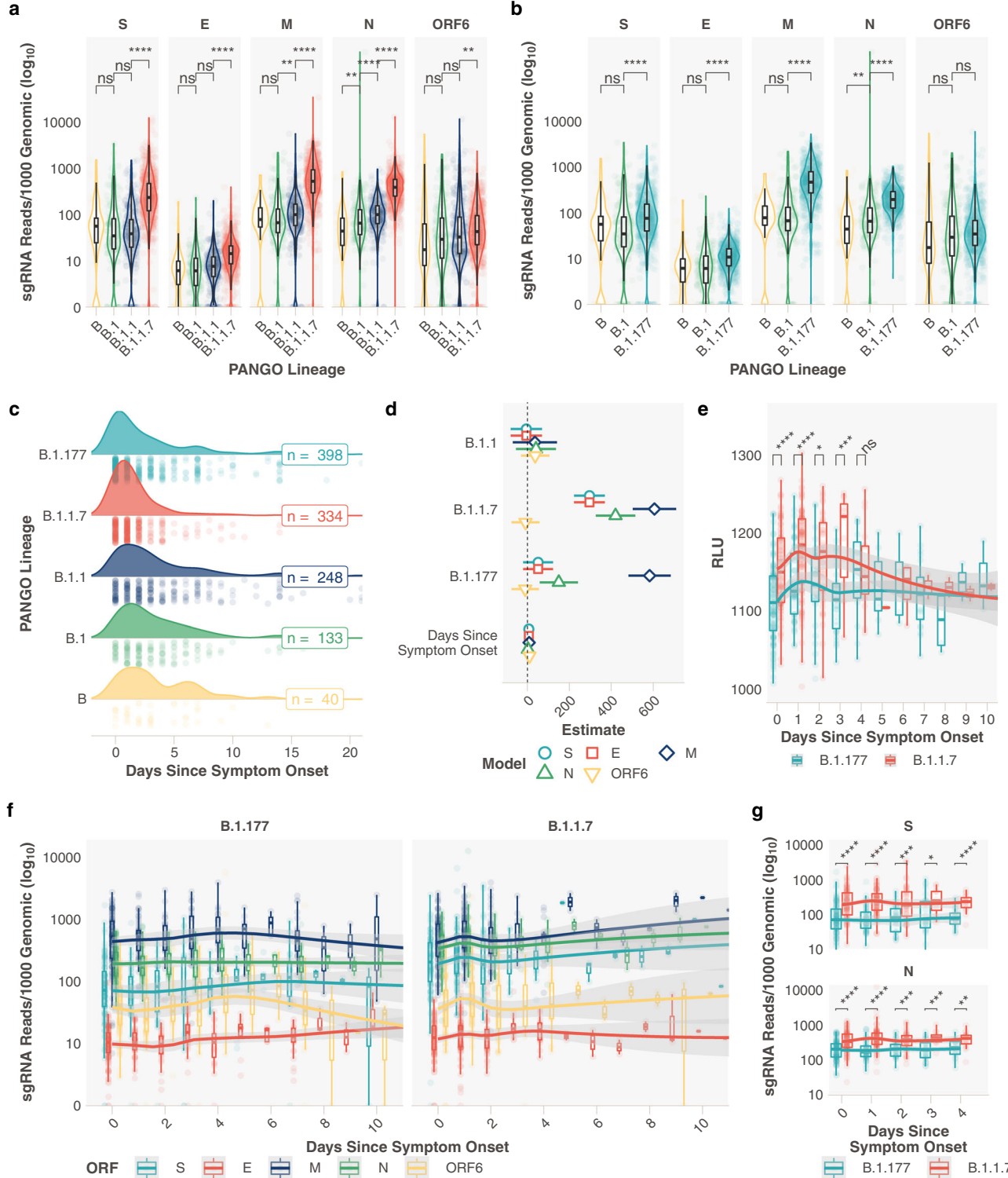

Protein measurements with sufficient replicates for statistical analysis were generated for N, M, S (Fig. 1j and Supplementary Data File 5) and Orf7a. All 4 proteins exhibited both a time and strain-dependent increase in the abundance of these proteins (N; B median 12 h = 14.6, B.1.1.7 median 12 h = 41.3, S; B median 12 h = 1.12, B.1.1.7 median 12 h = 2.67, M; B median 12 h = 0.631, B.1.1.7 median 12 h = 1.06).

To ensure the changes observed in clinical samples were not due to parental lineage changes in sgRNA abundance, such as those observed with the R203K/G204R mutation[15], we analysed

normalised sgRNA comparing B.1.1.7 and B.1.177 with their respective ancestral lineages (Fig. 2a, b). An increase in sgRNA is still observed for B.1.1.7 compared to B.1.1 in all major ORFs and interestingly, for B.1.177 compared to B.1 for ORFs S, E, M and N (Fig. 2a, b). This demonstrates the importance of stratifying sequences by lineage when studying sgRNA.

**Changes in subgenomic RNA abundance during the course of infection**. It is possible that the days since symptom onset at

**Fig. 2 Relationship between the day of symptom onset at sampling and subgenomic RNA levels in B.1.1.7 and B.1.177 infections. a** Subgenomic RNA data from most abundant ORFs for B.1.1.7 and respective ancestral lineages in Pillar 1 data ($B_n = 51$, $B.1_n = 164$, $B.1.1_n = 302$, $B.1.1.7_n = 729$). **b** Subgenomic RNA data from most highly expressed ORFs for B.1.177 and respective ancestral lineages in Pillar 1 data ($B_n = 51$, $B.1_n = 164$, $B.1.177_n = 764$). **c** Distribution of the reported days since symptom at the time of sampling for B ($n = 40$), B.1 ($n = 133$), B.1.1 ($n = 248$), B.1.177 ($n = 398$) and B.1.1.7 ($n = 334$) lineage infections. **d** Model estimates and 95% confidence intervals from generalised linear model comparing the effect of lineage and day of symptom onset on sgRNA abundance for most highly abundant ORFs. Reference lineage B.1. No interaction was observed between any lineage and the day of symptom onset. One B.1 sample with outlier N abundance (SHEF-CC370) was excluded. Error bars represent the standard error of the estimate. **e** Pseudo time course of RLU values for B.1.1.7 and B.1.177, plot restricted to 10 days. (0 Days; $B.1.177_n = 92$, $B.1.1.7_n = 106$. 1 Day; $B.1.177_n = 52$, $B.1.1.7_n = 102$. 2 Days; $B.1.177_n = 28$, $B.1.1.7_n = 43$. 3 Days; $B.1.177_n = 31$, $B.1.1.7_n = 14$. 4 Days; $B.1.177_n = 17$, $B.1.1.7_n = 19$.). **f** Normalised subgenomic RNA abundance for each of the most abundant ORFs stratified by the reported days since symptom onset and lineage, loess model. The plot is restricted to 10 days. (0 Days; $B.1.177_n = 115$, $B.1.1.7_n = 93$. 1 Day; $B.1.177_n = 53$, $B.1.1.7_n = 93$. 2 Days; $B.1.177_n = 31$, $B.1.1.7_n = 47$. 3 Days; $B.1.177_n = 31$, $B.1.1.7_n = 12$. 4 Days; $B.1.177_n = 19$, $B.1.1.7_n = 18$.). **g** Normalised sgRNA abundance for N and S is significantly increased at 0-4 days of infection. All *p* values calculated using an unpaired Wilcoxon signed-rank test, and adjusted for multiple testing with Holm (**** <0.0001, *** <0.001, ** <0.01, * <0.05, ns or blank = not significant). All boxplots depict the 25th, 50th (median), and 75th percentiles, and whiskers represent the most extreme datapoint which is no more than 1.5x the interquartile range. B.1.1.7 is represented in red, B.1.177 in teal, B in yellow, B.1 in green, B.1.1 in navy blue, B.1.1.1 in purple and other lineages in grey apart from D, S subgenomic RNA is teal, E is red, M is navy blue, N in green and ORF6 in yellow.

sampling may vary between lineages in our dataset, either due to changes in sampling practice over time or presentation of individuals to healthcare services, which in turn could confound our sgRNA findings. We obtained information on symptom duration at sampling for 2327 samples in our pillar 1 genome sequenced cohort (Supplementary Data File 6. A modest difference in the distribution of the days since symptom onset between B.1.177 and B.1.1.7 was apparent, where B.1.1.7 infections appeared to be sampled earlier (Fig. 2c). We therefore applied a generalized linear regression model to adjust for any effect this change in distribution might have on sgRNA abundance in lineages B.1, B.1.1, B.1.177 and B.1.1.7. Subgenomic RNA was still significantly elevated in B.1.1.7 in S, N, M, and E (Fig. 2d, Supplementary Fig. 5, and Supplementary Tables 3, 4). A smaller increase in sgRNA in B.1.177 (compared to B.1) was still seen, though only for M and N. Furthermore, sampling practice and sequencing in healthcare workers in our hospital remained consistent throughout the period when B.1.177 and B.1.1.7 were circulating, we hypothesised that these data were less likely to be affected by unknown confounders in sampling. Subgenomic RNA levels remained significantly higher in B.1.1.7 compared to B.1.177 infections when the comparison was restricted to samples from healthcare workers (Supplementary Fig. 6). No difference in sgRNA levels for B.1.1.7 was seen by sex or age (Supplementary Figs 7, 8). Relative light units in the diagnostic assay also varied over time in B.1.1.7 compared to B.1.177 infections, with significantly higher RLU at 0, 1, 2 and 3 days since symptom onset (Fig. 2e). Data since symptom onset for samples with Ct values were insufficient to replicate this analysis.

From these cross-sectional data, we created a pseudo infection time course using normalised sgRNA abundance on each day following symptom onset between B.1.1.7 and B.1.177 infections. This was made up of samples collected from different individuals on different days, plotted to represent a unitary time-series. A significantly higher total sgRNA for B.1.1.7 compared to B.1.177 was observed on day 1 following symptom onset (Supplementary Fig. 9a, *p* < 0.0001). In both lineages, M sgRNA followed by S and N sgRNA were the most abundant at each timepoint following symptom onset, with the peak at day 1 seen in B.1.1.7 infections for most ORFs (Fig. 2f). The increase in sgRNA for B.1.1.7 early in infection was most evident for S and N sgRNA, with significant differences found between B.1.1.7 and B.1.177 at days 0 to 4 (Fig. 2g). Less marked differences were seen for E and ORF6 sgRNA, with no difference in M sgRNA or other ORFs (Supplementary Fig. 9b, c). Although ORF6 has a significant increase on day 1 of infection (Supplementary Fig. 9b) and a clear shift in the peak of expression is evident (Fig. 2f - yellow). Using

the same methods for sgRNA estimation, we have previously shown that the presence of *ACE2* and *TMPRSS2* alter the kinetics of sgRNA expression in vitro, leading to a peak in abundance at an earlier stage of infection[9]. It is possible that the increased affinity for *ACE2* conferred by N501Y in the B.1.1.7 spike protein[16,17] enhances sgRNA profiles early in clinical infection. Alternatively, this may be a consequence of greater viral replication in B.1.1.7 due to alternative mechanisms.

**Increased prevalence of a noncanonical subgenomic RNA in B.1.1.7 infections**. Noncanonical sgRNA are the result of RdRp template switching from regions of the genome with no canonical TRS-B site. These noncanonical sgRNA tend to be enriched around canonical sites, presumably due to the frequency of RdRp template switching that occurs in close proximity. We quantified the abundance of these noncanonical sgRNA in pillar 1 B.1.1.7 infections (Supplementary Data File 7). We observed low levels of these transcripts throughout the genome across all samples[9] and compared the presence of noncanonical sgRNA transcripts between B.1.1.7 and B.1.177 infections (Fig. 3a). In addition to the noncanonical sgRNA resulting from the nucleocapsid R203K/G204R mutation (N*, Supplementary Fig. 2)[15], B.1.1.7 samples were also significantly enriched for reads which support the production of a noncanonical sgRNA from genomic position 28282 (Fig. 3b–d and Supplementary Fig. 9, Wilcoxon unpaired $p = <2e-16$, Wilcoxon effect size = 0.797, $B.1.1.7_n = 717$, $other_n = 430$). Like other sgRNAs, the production of this noncanonical sgRNA peaks 1 day following the onset of symptoms (Fig. 3e). The B.1.1.7 genomic sequence contains a triple nucleotide mutation at 28280, 28281, and 28282 GAT > CTA, resulting in an amino acid substitution, D3L, in the nucleocapsid protein. This triplet results in enhanced complementarity between the 28282 genomic regions and the sequence 3′ of the leader at the 5′ end of the genome (Fig. 3g panel iii), which may have resulted in a novel TRS-B-like site that drives higher sgRNA production from this locus. Of note, this site is upstream of the ORF9b ATG and this noncanonical sgRNA retains the full coding region of the ORF9b but lacks the canonical start codon of N. We propose that this represents the ORF9b sgRNA, which was detected at low levels in 13.8% (430/3127) non-B.1.1.7 samples (median normalised abundance=0.32), but is found in almost all pillar 1 B.1.1.7 infections (98.4%, 717/729), with a 16-fold increase in abundance (median normalised abundance = 6.02, Fig. 3a, c).

Interestingly, we noticed non-B.1.1.7 samples in our dataset with low levels of putative ORF9b sgRNA did not contain the

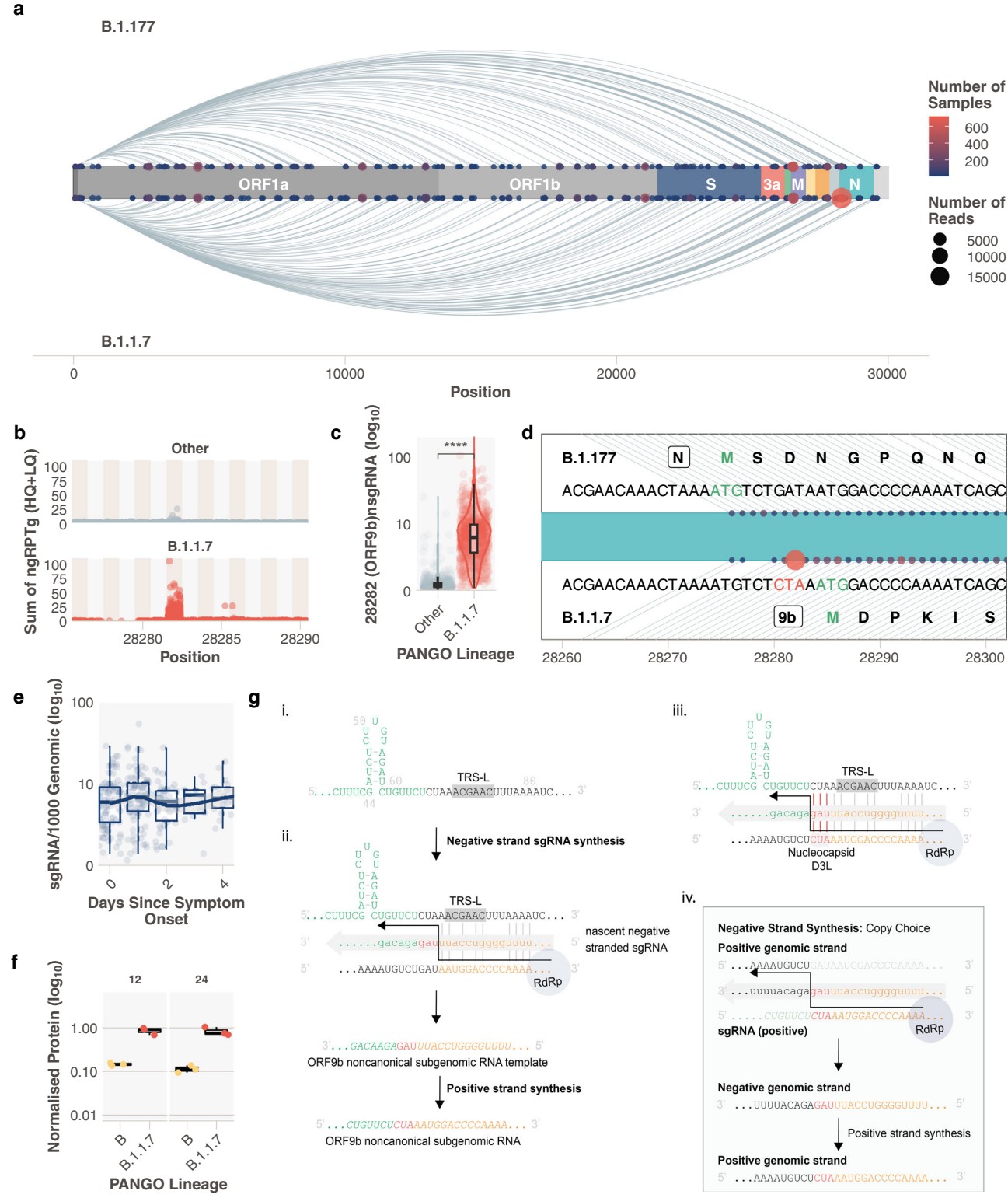

nucleocapsid D3L mutation. We propose the following hypothesis to explain our findings: in non-B.1.1.7 SARS-CoV-2, the weak complementarity around the ORF9b ATG causes low levels of noncanonical sgRNA to be produced which has high complementarity to the genome, but crucially contains the CTA triplet (Fig. 3g-ii and Supplementary Fig. 10a). This could have led to a transcriptionally driven recombination event, so-called copy choice, where RdRp uses the positive-stranded sgRNA as a template, and because of complementarity between the nascent

negative-strand switches templates to the positive-stranded genome after the CTA mutation, resulting in the GAT > CTA mutation in the genomic sequence (Fig. 3g-iv). Copy choice recombination between subgenomic RNA and the genomic RNA has previously been suggested as a method of recombination in viruses that employ discontinuous transcription[18]. This mutation creates a new TRS-B-like site with increased complementarity in B.1.1.7 viruses, driving higher sgRNA production from this locus, akin to sites of canonical sgRNA production (Fig. 3g-iii). A

**Fig. 3 A noncanonical sgRNA representing ORF9b is highly expressed in B.1.1.7 due to a triple nucleotide mutation in nucleocapsid leading to the D3L substitution. a** Subgenomic RNA not attributed to a canonical TRS-B site in B.1.177 and B.1.1.7 SARS-CoV-2 infections (Pillar 1 data). Size of the point is the number of reads and the colour is the number of samples (N* excluded for clarity), where red is the highest number of samples and navy blue is the lowest. **b** Total normalised noncanonical sgRNA (ngRPTg) in B.1.1.7 vs other lineages. **c** B.1.1.7 has significantly increased the abundance of a noncanonical sgRNA at 28282. (Wilcoxon effect size, unpaired = 0.797, $Other_n$ = 430, $B.1.1.7_n$ = 717). **d** Schematic of the noncanonical sgRNA (points) in the context of the SARS-CoV-2 genome around position 28282. The top shows the sequence present in B.1.177 and the amino acid sequence of the N protein. The bottom shows the sequence present in B.1.1.7 with the triplet CTA mutation and the closest ATG which represents the ORF9b methionine. **e** Noncanonical sgRNA at 28282 pseudo time course. **f** ORF9b protein levels measured using LC-MS/MS in B vs B.1.1.7 infected TMPRSS2 & ACE2 expressing A549 cells at 12 and 24 h post infection (n = 3 independent experiments). **g** Proposed model for sgRNA driven mutation of N D3L which leads to high ORF9b sgRNA abundance in B.1.1.7. All p values calculated using an unpaired Wilcoxon signed-rank test, and adjusted for multiple testing with holm (**** <0.0001, *** <0.001, ** <0.01, * <0.05, ns = not significant). All boxplots depict the 25th, 50th (median), and 75th percentiles, whiskers represent the most extreme datapoint which is no more than 1.5x the interquartile range. B.1.1.7 is represented in red, other lineages in grey and B in yellow.

similar event has been described previously more than 3′ in the N ORF[15], and mutations which result in increased complementarity and noncanonical sgRNA formation have been shown to occur in coronaviruses[19]. This raises the possibility that an ORF9b protein could be produced independently of other upstream sgRNAs with greater efficiency in B.1.1.7. ORF9b has been shown to regulate interferon responses[20–22] and has been shown to be present in SARS-CoV[23] and SARS-CoV-2 infections[11,23,24]. Interrogating our protein abundance data (Supplementary Data File 5) for ORF9b revealed a 6.3-fold increase in the median abundance of the ORF9b protein in B.1.1.7 after 12 h (n = 3, B normalised median = 0.145, B.1.1.7 normalised median = 0.914) and 6.7-fold increase at 24 h (n = 3, B normalised median = 0.121, B.1.1.7 normalised median = 0.749) (Fig. 3f). These findings are supported by another study showing both increased ORF9b subgenomic RNA, and higher protein levels of ORF9b (among others) in B.1.1.7 infections[25].

Resistance to Interferon appears to be significantly increased in B.1.1.7 infections when compared to all other lineages, including variants of concern like B.1.351/beta[26]. Our findings could therefore be important for investigating the greater transmissibility of B.1.1.7 and the role of D3L, in particular, requires urgent experimental validation. We also considered the possibility that this new TRS-B-like site could cause reduced transcription from upstream TRS sites due to early detachment of the transcription complex. The increased abundance of upstream subgenomic RNAs in B.1.1.7 SARS-CoV-2 suggests that this is an unlikely net consequence of this new TRS-B-like site. We explored whether the nucleocapsid D3L mutation has occurred in non-B.1.1.7 lineages in the global COG-UK sequence data set (14 February 2021). Seventy-seven sequences with D3L were noted in a variety of SARS-CoV-2 lineages and appear as homoplasies in a global phylogeny, suggesting that this event may have occurred independently on several occasions (Supplementary Fig. 11).

## Conclusions

Taken together, our data suggests that sgRNA abundance measurements from existing ARTIC Nanopore sequencing data can be used in real-time to examine the effect of SARS-CoV-2 variation on its ability to express its genome. These changes in expression are often represented in the amount of protein produced for these open reading frames, likely changing the phenotype of the virus. We cannot say if the increased sgRNA abundance is the cause or consequence of an increase in viral replication or a more efficacious entry, but our study provides further insight to guide exploration with mechanistic studies. A major advantage of this approach is that we can deconvolute the contribution of genomic and subgenomic RNA, which is impossible with current diagnostic PCR assays and we can, additionally, examine all ORFs simultaneously and discover noncanonical subgenomic RNA which could be of biological

relevance. Finally, we believe that sgRNA abundance analysis should be carried out on all compatible genomic surveillance platforms to give an instant readout of altered abundance in emerging SARS-CoV-2 variants. This would use existing data to complement epidemiological and phylodynamic methods, and provide an early warning of variants that might be of concern with regard to greater transmissibility and/or disease severity.

## Methods

**Diagnostic SARS-CoV-2 Testing.** SARS-CoV-2 positivity was determined from nose/throat swabs diagnostically by Sheffield Teaching Hospitals NHS Foundation trust either using the Hologic Aptima™ SARS-CoV-2 assay (Panther Fusion System) to generate relative light units (RLU)[27] or an in house dual E/RdRp real-time PCR assay to generate a cycle threshold (ECT or RCT respectively)[14]. The latter used the following primers and probes: RdRp gene F (GTGTGARATGGTC ATGTGTGGCGG), RdRp gene R (CARATGTTAAASACACTATTAGCATA), RdRp gene P (6-FAM- CAGGTGGAACCTCATCAGGAGATGC- BHQ1), E gene F (ACAGGTACGTTAATAGTTAATAGCGT), E gene R (ATATTGCAGCAGT ACGCACAC A) and E gene P (HEX-ACACTAGCCATCCTTACTGCGCTTCG-BHQ1). Nucleic acid was extracted on the MagnaPure96 platform (Roche Diagnostics Ltd, UK) and 6 µL of extract used to amplify E and RdRP genes on an ABI Thermal Cycler (Applied Biosystems, USA).

**Sample preparation, ARTIC network PCR and nanopore sequencing.** RNA was extracted from viral transport medium (VTM) with Qiagen QIAamp MinElute Virus Spin Kit (50). Resultant RNA was subject to the ARTIC network[8] tiled amplicon protocol (Supplementary Data File 8) and subsequently sequenced on an Oxford Nanopore Technologies GridION X5. Bases were called with the default basecaller in MinKNOW (currently guppy v4) with —require-both-ends set for demultiplexing. Raw sequencing fastq files were subject to no processing after sequencing to ensure sgRNA leader sequences are retained.

**Subgenomic read classification and normalisation.** Subgenomic RNAs were classified using periscope[9] v0.0.8a. Briefly, reads containing the leader sequence at their start are identified by a local alignment, the quality of this alignment determines which quality bin sgRNA reads are placed (high quality—HQ, low quality—LQ or low low quality—LLQ). These bins are determined as follows: HQ; local alignment score >50 and at known ORF site (HQ canonical sgRNA) or not within ORF or primer site (HQ noncanonical sgRNA). LQ; local alignment score <50 but >30 and at known ORF (LQ canonical sgRNA) or not within ORF or primer site (LQ noncanonical sgRNA). LLQ; local alignment score <30 but at known ORF

For normalisation, the amplicon from which the sgRNA evidence was generated is determined and a count of genomic reads for this amplicon is used to normalise the raw sgRNA read counts.

Samples were excluded from the subsequent analysis if their consensus coverage was <0.9 and/or they had less than 50,000 mapped reads (we have previously shown that fewer reads produce a less robust analysis).

**Lineage assignment.** Lineages[1] were assigned using Pangolin (https://github.com/cov-lineages/pangolin) v2.1.7).

**Viral culture**

*Cells and virus isolates.* TMPRSS2 & ACE2 expressing A549 cells[28] were cultured under standard conditions (37 °C and 5% $CO_2$) in F-12K medium (Thermo Fisher Scientific) supplemented with 10% heat-inactivated foetal bovine serum (Gibco), 100 units/mL penicillin, 100 µg/mL streptomycin, 2 mM L-glutamine (Gibco), 25 mM HEPES, 1% non-essential amino acids (Gibco), 2 µg/mL hygromycin B and 2 mg/mL G418. All cells (for viral passage, plaque assay and infections) were

routinely tested and confirmed to be free of mycoplasma (MycoAlert Mycoplasma Detection Kit (Lonza).

The BetaCoV/Australia/VIC01/2020 strain of SARS-CoV-2 (PANGO lineage B) was obtained from the Victorian Infectious Diseases Reference Laboratory, Melbourne through Public Health England (Colindale), in April 2020. The BetaCoV/England/MIG457/2020 (B.1.1.7) strain of SARS-CoV-2 (B.1.1.7) was obtained from Public Health England (Colindale), in January 2021. B was passaged once on Vero cells and B.1.1.7 was passaged once on VeroE6 + ACE2 + TMPRSS2 cells[28] to generate the stocks used in this study. For these virus propagations, cells were infected (MOI = 0.01) for 1 h at room temperature. The entire flask was frozen at 48 h post infection. Following three freeze-thaw cycles, debris was removed by clarification (centrifugation at 2500 rpm, 10 min) and viral stocks were aliquoted and stored at 80 °C. Viral titres were calculated in plaque-forming units (PFU/mL) by standard plaque assay[29,30]. Virus sequences were verified[31], and no additional mutations were identified in the stocks compared to the parental sequences. All lineage-defining mutations were confirmed to be present.

*Infections.* The day before infection, A549 + ACE2 + TMPRSS2 cells were seeded at 60% confluency in six-well plates. Cells were washed once with PBS before incubation with virus diluted in sera-free media (MOI = 1), for 1 h at room temperature on a rocking platform. After the inoculum was removed, cells were washed with PBS and 2 mL of media containing 2% foetal bovine serum was added. At 12 and 24 h post infection, cell monolayers were harvested in 250 μL of 2x Laemlli buffer (for protein lysate samples).

*Quantitative mass spectrometry analysis.* About 20 ml of Laemmli buffer cell lysates from infected cells were alkylated with 25 mM iodoacetamide in the dark for 30 min at 37 °C. TEAB was added to a final concentration of 50 mM, and protein was trapped and washed on S-trap micro spin columns (ProtiFi, LLC) according to the manufacturer's instructions. Protein was digested using 5 mg trypsin sequence grade (Pierce) at 47 °C for 1 h and 37 °C for 1 h. Eluted peptides were dried in a vacuum concentrator and resuspended in 40 ml 0.5% formic acid for LC-MS/MS analysis. Peptides were analysed using nanoflow LC-MS/MS using an Orbitrap Elite (Thermo Fisher) hybrid mass spectrometer equipped with a nanospray source, coupled to an Ultimate RSLCnano LC System (Dionex). Peptides were desalted online using a nano trap column, 75 μm I.D.X 20 mm (Thermo Fisher) and then separated using a 130-min gradient from 3 to 35% buffer B (0.5% formic acid in 80% acetonitrile) on an EASY-Spray column, 50 cm × 50 μm ID, PepMap C18, 2 μm particles, 100 Å pore size (Thermo Fisher). The Orbitrap Elite was operated with a cycle of one MS (in the Orbitrap) acquired at a resolution of 120,000 at m/z 400, with the top 20 most abundant multiply charged (2+ and higher) ions in a given chromatographic window subjected to MS/MS fragmentation in the linear ion trap. An FTMS target value of 1e6 and an ion trap MSn target value of 1e4 were used with the lock mass (445.120025) enabled. Maximum FTMS scan accumulation time of 500 ms and maximum ion trap MSn scan accumulation time of 100 ms were used. Dynamic exclusion was enabled with a repeat duration of 45 s with an exclusion list of 500 and an exclusion duration of 30 s. Raw mass spectrometry data were analysed with MaxQuant version 1.6.10.43[32]. Data were searched against a combined sequence database including the Human and SARS-CoV-2 UniProt reference proteomes using the following search parameters: digestion set to Trypsin/P, methionine oxidation and N-terminal protein acetylation as variable modifications, cysteine carbamidomethylation as a fixed modification, match between runs enabled with a match time window of 0.7 min and a 20-min alignment time window, label-free quantification (LFQ) was enabled with a minimum ratio count of 2, the minimum number of neighbours of 3 and an average number of neighbours of 6. A protein FDR of 0.01 and a peptide FDR of 0.01 were used for identification level cut-offs based on a decoy database searching strategy. SARS-CoV-2 protein identification and quantification data were extracted and normalised to the levels of proteins encoded by the ORF1ab gene. Nsp1, Nsp2 and Nsp3 were quantified across all replicate samples at each time point, and strain and, therefore, the summed intensity of these proteins were used as a normalisation factor for other SARS-CoV-2 proteins.

**Statistics and reproducibility.** Statistical analysis was performed in R[33]. Figures were generated in R using tidyverse[34] (Supplementary Data File 9), apart from those that depict sequencing reads, which were generated in IGV[35]. Tests between groups were performed using an unpaired Wilcoxon test using the rstatix package (https://github.com/kassambara/rstatix), adjusting *p* values for any multiple comparisons using the Holm method. All boxplots shown have median as the centre line, upper and lower lines as 75th and 25th percentile respectively, outliers are not specifically shown as all points are represented. Whiskers represent 1.5x interquartile range.

**Phylogenetic tree generation.** The grapevine pipeline (https://github.com/COG-UK/grapevine) was used for generating the phylogeny based on all data available on GISAID and COG-UK up until 16 February 2021. A representative sample of global sequences was obtained in two steps. First by randomly selecting one sequence per country per epi week followed by random sampling of the remaining sequences to generate a sample of 4000 sequences. The global tree was then pruned using code adapted from the tree-manip package (https://github.com/josephhughes/tree-manip). We then identified samples with D3L mutations and colour coded these tips according to their lineages. The visualisation was produced using R/ape, R/ggplot2, R/ggtree, R/treeio, R/phangorn, R/stringr, R/dplyr, R/aplot.

**Ethics approval and consent.** Individuals presenting with active COVID-19 disease were sampled for SARS-CoV-2 sequencing at Sheffield Teaching Hospitals NHS Foundation Trust, UK using samples collected for routine clinical diagnostic use. This work was performed under approval by the Public Health England Research Ethics and Governance Group for the COVID-19 Genomics UK consortium (R&D NR0195). Approval was provided to undertake viral sequencing on residual clinical diagnostic samples and analysis of anonymised data without the individual patients' consent.

**Reporting summary.** Further information on research design is available in the Nature Research Reporting Summary linked to this article.

## Data availability

All SARS-CoV-2 consensus sequences that are of high enough quality are available on GISAID and ENA and from https://www.cogconsortium.uk/data/. All sgRNA abundance data were provided as supplementary data at https://github.com/sheffield-bioinformatics-core/periscope-variants-publication. All raw sequencing data are available on ENA under the accession PRJEB48895. The mass spectrometry proteomics data has been deposited to the ProteomeXchange Consortium via the PRIDE partner repository with the dataset identifier PXD029954. All supplementary data files.

## Code availability

Periscope is available at https://github.com/sheffield-bioinformatics-core/periscope and version 0.0.8a used in this publication is available at https://doi.org/10.5281/zenodo.5717129 the code used to generate the figures contained within this manuscript can be found as a supplementary data, at https://github.com/sheffield-bioinformatics-core/periscope-variants-publication or also available at https://doi.org/10.5281/zenodo.5717129.

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

## Acknowledgements

We thank the Sheffield Bioinformatics Core for their useful thoughts and discussions. We would like to thank the members of the Sheffield Biomedical Research Centre for their continued support of the SARS-CoV-2 sequencing work in Sheffield. We also thank Public Health England and the Victorian Infectious Diseases Reference Laboratory, Melbourne, for providing virus isolates. The BetaCoV/Australia/VIC01/2020 strain of SARS-CoV-2 can be provided by the Victorian Infectious Diseases Reference Laboratory, Melbourne, through Public Health England pending scientific review and a completed material transfer agreement. The SARS-CoV-2 VOC B.1.1.7 isolate can be provided by Public Health England pending scientific review and a completed material transfer agreement. Sequencing of SARS-CoV-2 samples was undertaken by the Sheffield COVID-19 Genomics Group as part of the COG-UK CONSORTIUM and supported by funding from the Medical Research Council (MRC) part of UK Research & Innovation (UKRI), the National Institute of Health Research (NIHR) and Genome Research Limited, operating as the Wellcome Sanger Institute. M.D.P. and D.W. are funded by the NIHR Sheffield Biomedical Research Centre (BRC - IS-BRC-1215-20017). T.I.d.S. is supported by a Wellcome Trust Intermediate Clinical Fellowship (110058/Z/15/Z). J.R.E. is supported by a Sir Henry Dale Fellowship jointly funded by the Wellcome Trust and the Royal Society (216370/Z/19/Z). A.A.P. is supported by the BBSRC (BB/S009566/1). H.S. is supported by Wellcome Trust (106207) and European Research Council (646891). We thank all partners of and contributors to the COG-UK consortium, who are listed at https://www.cogconsortium.uk/about/.

## Author contributions

Conceptualisation; M.D.P., T.I.d.S., A.A.P., H.S., M.O.C., S.L., S.G. and S.M. Data curation; M.D.P., T.I.d.S., B.B.L., D.R.S., A.J.K., D.G.P., S.E.H., M.R., C.E., O.M.S. and H.A. Formal analysis; M.D.P., T.I.d.S., B.B.L., S.E.H. and A.A.P. Funding acquisition; D.W., T.I.d.S., J.R.E. and A.E.F. Investigation; P.Z., M.G., S.F.L., H.R.H., M.W., B.H.F., S.C., P.W., M.P., L.R.G., H.S., A.P.P., O.M.S., H.A. and M.O.C. Methodology; M.D.P. and T.I.d.S. Project administration; N.S. and S.E.H. Resources; C.E., M.R., D.G.P., A.C., A.S., K.J., N.A., R.R., H.S., A.A.P., O.M.S. and H.A. Software; M.D.P. and J.H. Supervision; M.D.P., D.W., T.I.d.S., J.R.E. and A.E.F. Validation; H.S., O.M.S., M.O.C., H.A. and A.A.P. Visualisation; M.D.P., B.B.L., S.H. and A.A.P. Writing—original draft; M.D.P., T.I.d.S., H.S. and A.A.P. Writing—review & editing; M.D.P., D.W., B.B.L., S.H., T.I.d.S., H.S. and A.A.P.

## Competing interests

Mark O. Collins is an Editorial Board Member for Communications Biology but was not involved in the editorial review of, nor the decision to publish this article. Since this manuscript was accepted for publication Matthew Parker is now an employee of Oxford Nanopore Technologies and the remaining authors declare no competing interests.

## Additional information

## The COVID-19 Genomics UK (COG-UK) consortium

Matthew D. Parker [1,2], Benjamin B. Lindsey [5,6], Sharon Hsu[2,6], Alexander J. Keeley [5,6], David G. Partridge [5], Amy State[5], Joe Heffer [8], Nikki Smith[6], Peijun Zhang[6], Marta Gallis[6], Stavroula F. Louka [6], Hailey R. Hornsby [6], Max Whiteley[6], Benjamin H. Foulkes[6], Stella Christou [6], Paige Wolverson [6], Manoj Pohare [6], Samantha E. Hansford [6], Luke R. Green [6], Cariad Evans[5], Mohammad Raza[5] & Thushan I. de Silva [5,6 ✉]

A full list of members and their affiliations appear in the Supplementary Information.

