## [Peer Review File · Communications Biology]

REVIEWERS' COMMENTS:

Reviewer #1 (Remarks to the Author):

In the manuscript by Parker et al, they analysed sequencing data from 4400 SARS-CoV-2 positive clinical samples at Sheffield Teaching Hospitals and showed that normalised subgenomic mRNA (sgRNA) is significantly increased in the alpha variant (B.1.1.7) compared to the previous dominant circulating lineage in the UK (B.1.177). Furthermore, they described that the non-canonical ORF9 sgRNA was found in 98.4% of B.1.1.7 SARS-CoV-2 infections compared with only 13.8% of other lineages. Finally, from these observations, they hypothesized that differences in sgRNA abundance may be related to differences in viral phenotypes among variants, including viral transmission and disease severity.

In general, most experiments are well controlled and convincing, the paper is well written and the results are presented in a clear and comprehensible manner.

Overall, most critiques raised by reviewer #3 have been adequately addressed and the text has been modified accordingly.

Reviewer #2 (Remarks to the Author):

The study by Matthew D Parker et al use long-read sequencing dataset in order to examine the SARS-CoV-2 lineage B.1.1.7 transmission capacity. The study provides an interesting, important hypothesis that can form the basis of future research projects. The analysis carried out rigorously, the improvements which were carried out based on the original reviews resulted in a manuscript which is suitable for publication in Communications Biology.